# Hepatic arterial infusion is effective in patients with unresectable colorectal liver metastases refractory to standard systemic chemotherapy: A retrospective cohort study

Masatsugu Ishii[1,2], Osamu Itano[3]*, Hideki Iwamoto[1,4], Yuko Takami[5], Naomi Okada[6], Tetsuya Inoue[7], Satoshi Itano[1]

1 Department of Gastroenterology, Kurume Chuo Hospital, Kurume, Japan, 2 Department of Hepato-Biliary-Pancreatic Surgery, Tochigi Cancer Center, Tochigi, Japan, 3 Department of Hepato-Biliary-Pancreatic & Gastrointestinal Surgery, School of Medicine, International University of Health and Welfare, Chiba, Japan, 4 Division of Gastroenterology, Department of Medicine, Kurume University School of Medicine, Kurume, Japan, 5 Department of Hepato-Biliary-Pancreatic Surgery and Clinical Research Institute, National Hospital Organization Kyushu Medical Center, Kyushu, Japan, 6 Department of Oncology, Naomi Clinic, Tokyo, Japan, 7 Department of Surgery, Hita Central Hospital. Hita, Japan

* itano@ihwg.jp

## Abstract

We identified an effective chemotherapy regimen in patients refractory to standard chemotherapy. We included patients with unresectable colorectal liver metastases who underwent hepatic artery infusion chemotherapy and systemic chemotherapy between January 2015 and December 2022. This study was a retrospective analysis conducted at a single center. The patients received either biweekly oxaliplatin and 5-fluorouracil through hepatic artery infusion chemotherapy as well as bevacizumab and leucovorin injected intravenously (HAIC-FOLFOX-B) or biweekly irinotecan and 5-fluorouracil by hepatic artery infusion chemotherapy and bevacizumab and leucovorin injected intravenously (HAIC-FOLFIRI-B). Of the 42 patients, 20 underwent HAIC-FOLFOX-B while 22 underwent HAIC-FOLFIRI-B treatment with response rates of 25% and 4.5%, respectively. The median overall survival and progression-free survival were 12.9 and 4.7 months and 17.4 and 7.7 months in patients undergoing HAIC-FOLFOX-B and HAIC-FOLFIRI-B, respectively. The overall incidence of grade 3/4 toxicity was 23.8%. However, no treatment-related deaths occurred. Functional catheter-associated problems occurred in 9.5% of the patients. Hepatic arterial occlusion occurred in three patients (7.1%); catheter-associated infection occurred in one (2.4%) patient. However, these occurrences were not life-threatening complications. HAIC-FOLFOX-B and HAIC-FOLFIRI-B might improve survival in patients with unresectable colorectal liver metastases and in those who underwent both systemic oxaliplatin-based and irinotecan-based chemotherapies and were refractory to them. HAIC FOLFOX-B and FOLFIRI-B regimens might be

**Data availability statement:** All relevant data are within the manuscript and its Supporting Information files.

**Funding:** The author(s) received no specific funding for this work.

**Competing interests:** The authors have declared that no competing interests exist.

effective therapeutic options in patients with unresectable colorectal liver metastases refractory to standard systemic chemotherapy.

## Introduction

Colorectal cancer (CRC) is one of the most common cancers and the most common cause of cancer-related deaths worldwide. At least 50% of patients with CRC develop metastases [1]. The liver is the most frequent site of metastasis from CRC. Such metastatic tumors subsequently cause hepatic dysfunction and failure and eventually death. In cases of advanced CRC, a combination treatment using three cytotoxic agents, including 5-fluorouracil, oxaliplatin, and irinotecan, and two molecular-targeted agents, including bevacizumab and epidermal growth factor receptor antibody (cetuximab or panitumumab), is the standard systemic therapy worldwide [2–10]. These approaches have improved the response and survival rates in patients with advanced CRC. However, an effective chemotherapy regimen in patients with advanced CRC that is refractory to these standard chemotherapies has not yet been established. Although regorafenib or trifluridine/tipiracil (TAS-102) is recommended, the survival benefits of these chemotherapies lasted 6.4–9.0 months, after which several adverse effects were encountered [11,12].

Many trials published in the 1990s that compared hepatic artery infusion (HAIC) alone with systemic chemotherapy showed superior response rates in favor of HAIC. However, HAIC failed to consistently show improvements in the overall survival (OS) rates [13–15]. In comparative studies on systematic treatment, HAIC did not yield any survival benefit; therefore, it is not often used nowadays [16,17]. However, these studies reported HAIC using only 5-fluorouracil and there are very few reports on HAIC using oxaliplatin plus 5-fluorouracil or irinotecan plus 5-fluorouracil and systemic chemotherapy using bevacizumab [18,19]. Therefore, this study aimed to assess the efficacy of HAIC using oxaliplatin or irinotecan; 5-fluorouracil; and systemic chemotherapy with bevacizumab in patients with unresectable metastatic liver lesions from CRC through a retrospective evaluation of patients who underwent these treatments at our hospital.

## Materials and methods

### Ethics statements

This study was approved by the Ethics Committee of Kurume Chuo Hospital (approval number: 20210002,20230001). This study was conducted in accordance with the principles of the Declaration of Helsinki. Written informed consent was obtained from each patient prior to participation.

### Patients

Overall, 42 patients were scheduled to receive HAIC at Kurume Chuo Hospital between January 2015 and December 2022. This hospital is known as a specialized HAIC facility. All patients were referred from other hospitals, that adhered to the

standard of care and were trustworthy. Forty-two patients with unresectable CRC liver metastasis refractory to standard systemic chemotherapy and who underwent HAIC using 5-fluorouracil and oxaliplatin or irinotecan and systemic chemo-therapy using bevacizumab between January 2015 and December 2022 were included in this study. Unresectability of hepatic metastases was assessed by the physician in charge for various reasons, including tumor progression, age, and performance status. Of the 42 patients included 42 had previously undergone both oxaliplatin-based and irinotecan-based chemotherapies. The inclusion criteria for HAIC were: Eastern Cooperative Oncology Group Performance Status of 0 or 1, age ≥ 20 years, adequate bone marrow function (white blood cell count ≥ 3,000/mm$^3$, neutrophil count ≥ 1,500/mm$^3$, platelets ≥ 100,000/mm$^3$, and hemoglobin level ≥ 8.0 g/dL), serum total bilirubin ≤ 3.0 mg/dL, adequate renal function (serum creatinine level ≤ 1.5 mg/dL, adequate oral intake, and creatinine clearance or estimated glomerular filtration rate according to the Cockcroft–Gault formula ≥ 60 mL/min), no abnormal findings on electrocardiography within 30 days since HAIC initiation, and provision of written informed consent. The exclusion criteria for this treatment were as follows: post-treatment follow-up period < 1 month (n = 2) and no history of chemotherapy (n = 7). Of the patients who had previously under-gone both oxaliplatin-based and irinotecan-based chemotherapies, 20 and 22 underwent HAIC-FOLFOX-B and HAIC-FOLFIRI-B, respectively ([Fig 1]). All of these patients were refractory to both oxaliplatin-based and irinotecan-based chemotherapies. Refractory patients were defined as those who could not receive either regimen again due to adverse effects or disease progression. The study followed the Strengthening the Reporting of Observational Studies in Epidemiol-ogy guidelines for observational studies [20].

## Data collection and treatment regimen

The hospital database was used to extract patient data; liver metastases were diagnosed based on either pathological findings or typical imaging. These data have been accessed for research purposes since February 2023. Computed tomography was performed during hepatic arteriography using a catheter in the hepatic artery. Arteries supplying blood to tumors were identified.

HAIC was performed with a delivery port system or new indwelling catheter system, (i.e., System i) [21,22]. The use of HAIC at our institution has been previously described [23,24]. We administered oxaliplatin (50 mg) for 2 h and 5-fluorouracil (1,250 mg) through 46-h continuous infusion via the artery. Bevacizumab (5 mg/kg) and leucovorin (200 mg/kg) were injected intravenously on day 1 (HAIC-FOLFOX-B). We administered irinotecan (60 mg) for 2 h and 5-fluorouracil (1250 mg) through 46-h continuous infusion via the artery as well as bevacizumab (5 mg/kg) and leucovorin (200 mg/m$^2$) intravenously on day 1 (HAIC-FOLFIRI-B). If multiple arteries were supplying blood to the tumor, oxaliplatin or irinote-can and 5-fluorouracil were administered through a different artery in the next course. We used these regimens once every 2 weeks. Our first choice HAIC was HAIC-FOLFOX-B, but this was decided by the physician in charge. Regarding

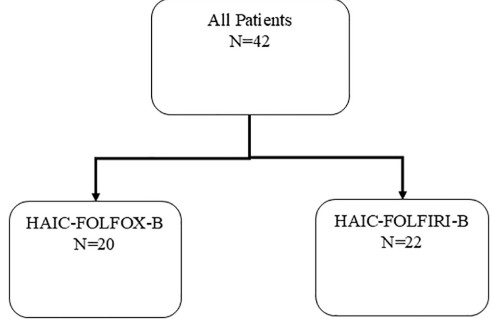

**Fig 1. Patient classification.**

patient allocation, the physician in charge considered the previously administered regimen and decided the regimen to be selected.

HAIC was performed with a delivery port system or new indwelling catheter system, i. e., System-i. At the start of each course, we replaced this catheter into the artery that supplied blood to tumor.

### Treatment evaluation

The primary endpoint was tumor response. The patients were monitored monthly for disease recurrence, and the overall assessment included physical examination, radiography, ultrasonography, computed tomography, and laboratory examinations. Tumor response to therapy was assessed according to the Response Evaluation Criteria in Solid Tumors version 1.1 using computed tomography or magnetic resonance imaging. The secondary endpoints were OS, progression-free survival (PFS) and adverse effects. The survival period was calculated as the period from the day of commencing this treatment to the day of all-cause mortality. Adverse drug reactions were assessed according to the Common Terminology Criteria for Adverse Events version 5.0.

### Statistical analysis

The Kaplan–Meier method was performed to estimate the time to progression and OS. The Mann–Whitney U test was performed to compare continuous variables. All statistical analyses were performed using Statistical Package for the Social Sciences (IBM Corp., Armonk, NY, USA). The level of statistical significance was set at a $P$-value of <0.05.

## Results

### Patient characteristics

Almost all patients presented with unresectability of hepatic metastases as assessed by the surgeons from other hospitals. All patients were identified to be refractory to intravenous chemotherapy according to tumor progression or the development of an adverse event.

Table 1 summarizes the patient characteristics; of the participants, there were 21 male and 21 female patients, (age ranges, 37–81 years). All patients had undergone systemic chemotherapy previously and were refractory to it. Synchronous liver metastases were observed in 36 patients (85.7%). Forty patients (95.2%) had multiple liver metastases (>10). Thirty-eight patients (90.5%) had undergone primary lesion resection. All patients had undergone previous treatment in the other hospital and these indications showed slight differences in each department. There were no differences in the background factors between patients who received HAIC-FOLFOX-B and those who received HAIC-FOLFIRI-B. Patients did not undergo hepatectomy after the aforementioned treatments for various reasons, including age, and performance status.

### Effect and survival

The median follow-up period for the patients was 9.7 (range, 1–59) months after HAIC treatment initiation. The patients received a median of 7.5 (range, 1–50) cycles. Table 2 presents the best response to HAIC. The response (complete response [CR] + partial response [PR]) and disease control rates (CR + PR + stable disease) were 25% and 70%, among the patients who underwent HAIC-FOLFOX-B, respectively. The median OS and PFS periods after treatment initiation were 12.9 months (95% confidence interval [CI], 4.892–20.908) and 4.7 months (95% CI, 2.977–6.423), respectively (Fig 2 A, B). The OS rates after disease onset were 59% at 3 years and 22.1% at 5 years. The median OS periods since disease onset was 45.9 months (95% CI, 28.43–63.37) (Fig 2 C). The response and disease control rates were 4.5% and 86.3%, respectively, among the patients who underwent HAIC-FOLFIRI-B, respectively. The median OS and PFS periods after treatment initiation were 17.4 months (95% CI, 7.351–27.515) and 7.7 months (95%

**Table 1. Patient characteristics.**

| | | All (N = 42) | HAIC-FOLFOX-B (n = 20) | HAIC-FOLFIRI-B (n = 22) | P-value |
|---|---|---|---|---|---|
| Age (years) | | 62.5 (37–80) | 62 (37–72) | 63.5 (40–80) | 0.588 |
| Sex | Male | 21 (50) | 11 (55) | 10 (45.5) | 0.537 |
| | Female | 21 (50) | 9 (45) | 12 (54.5) | |
| Location of primary cancer | Colon, right side | 10 (23.8) | 3 (15) | 7 (31.8) | 0.201 |
| | Colon, left side | 32 (76.2) | 17 (85) | 15 (68.2) | |
| Previous chemotherapy before our treatment | 2 | 9 (21.4) | 4 (20) | 5 (22.7) | 0.614 |
| | 3 | 14 (33.3) | 9 (45) | 5 (22.7) | |
| | 4 | 9 (21.4) | 4 (20) | 5 (22.7) | |
| | 5 | 7 (16.7) | 2 (10) | 5 (22.7) | |
| | 6 | 2 (4.8) | 1 (5) | 1 (4.5) | |
| | 7 | 1 (2.4) | 0 (0) | 1 (4.5) | |
| Previous anti-EGFR antibody | | 22 (52.3) | 11 (55) | 11 (50) | 0.746 |
| Tumors (n) | Unifocal | 2 (4.8) | 2 (10) | 0 (0) | 0.129 |
| | Multifocal | 40 (95.2) | 18 (90) | 22 (100) | |
| Maximum diameter (mm) | | 59.3 (14–190.9) | 73.8 (20.2–190.9) | 51.0 (14.0–123.7) | 0.307 |
| Alkaline phosphatase (IU/mL) | | 696 (55–3306) | 802.5 (186–2467) | 540.5 (55–3306) | 0.151 |
| CEA (ng/mL) | | 265.2 (6.2–14040.4) | 847 (6.5–12316.8) | 157.1 (6.2–14040.4) | 0.268 |
| CA19−9 (U/mL) | | 224.6 (2–22469) | 620.5 (2–22469) | 170.9 (2–11312) | 0.940 |
| Synchronous | Yes | 36 (85.7) | 17 (85) | 19 (86.3) | 0.900 |
| | No | 6 (14.3) | 3 (15) | 3 (13.6) | |
| Distant metastasis | Yes | 23 (54.8) | 11 (55) | 12 (54.5) | 0.976 |
| | No | 19 (45.2) | 9 (45) | 10 (45.5) | |
| Metastatic organ | Lung | 16 (38.1) | 7 (35) | 9 (40.9) | |
| | Lymph node | 4 (9.5) | 2 (10) | 2 (9.1) | |
| | Peritoneum | 5 (11.9) | 4 (20) | 1 (4.5) | |
| | Brain | 1 (2.4) | 0 (0) | 1 (4.5) | |
| Primary lesion resection | Yes | 38 (90.5) | 18 (90) | 20 (90.1) | 0.920 |
| | No | 4 (9.5) | 2 (10) | 2 (9.1) | |
| K-RAS | Wild type | 23 (54.8) | 12 (60) | 11 (50) | 0.725 |
| | Mutant | 13 (31) | 5 (25) | 8 (36.4) | |
| | Unknown | 6 (14.3) | 3 (15) | 3 (13.6) | |

Data are expressed as medians (ranges) or n (%). CEA, carcinoembryonic antigen; CA19−9, carbohydrate antigen 19−9; EGFR, epidermal growth factor receptor; HAIC-FOLFOX-B, hepatic artery infusion using 5-fluorouracil plus oxaliplatin combination therapy and systemic chemotherapy using leucovorin and bevacizumab; HAIC-FOLFIRI-B, hepatic artery infusion using 5-fluorouracil plus irinotecan combination therapy and systemic chemotherapy using leucovorin and bevacizumab; K-RAS, Kirsten rat sarcoma viral oncogene homolog.

CI, 4.178–11.156) (Fig 2 A, B), respectively. The OS rates after disease onset were 66.5% at 3 years and 42.2% at 5 years. The median OS periods after disease onset was 49.3 months (95% CI, 33.55–65.05) (Fig 2C). There was no significant difference in the OS and PFS rates after treatment initiation in patients who underwent HAIC-FOLFOX-B and

**Table 2. Response rate among the patients in this study.**

| Best response to HAIC | Previous systemic oxaliplatin-based and irinotecan-based chemotherapies. (N=42) | |
|---|---|---|
| | HAIC-FOLFOX-B (n=20) | HAIC-FOLFIRI-B (n=22) |
| Complete response | 0 (0) | 0 (0) |
| Partial response | 5 (25) | 1 (4.5) |
| Stable disease | 9 (45) | 18 (81.8) |
| Disease progression | 6 (30) | 3 (13.6) |
| Response rate (%) | 25 | 4.5 |
| Disease control rate (%) | 70 | 86.3 |

Data are expressed as n (%). HAIC-FOLFOX-B, hepatic artery infusion using 5-fluorouracil plus oxaliplatin combination therapy and systemic chemotherapy using leucovorin and bevacizumab; and HAIC-FOLFIRI-B, hepatic artery infusion using 5-fluorouracil plus irinotecan combination therapy and systemic chemotherapy with leucovorin and bevacizumab.

in those who underwent HAIC-FOLFIRI-B ($P=0.116$ and $P=0.053$, respectively) (Fig 2 A, B). There was also no significant difference in the OS rates after disease onset between patients who underwent HAIC-FOLFOX-B and those who underwent HAIC-FOLFIRI-B ($P=0.470$) (Fig 2 C).

## Adverse effects

Adverse effects were evaluated in 42 patients. Table 3 summarizes the adverse effects of the patients. Of these complications, grade 3 or 4 toxicities (defined according to the Common Terminology Criteria for Adverse Events version 5.0) were noted in seven patients with anemia (16.7%), two with allergies (4.8%), and one with liver infection (2.4%). The overall incidence of grade 3/4 toxicity was 23.8% (10/42). No treatment-related deaths occurred. Functional catheter-associated problems occurred in 9.5% (4/42) of patients. Hepatic artery occlusion occurred in three patients (7.1%) and catheter-associated infection occurred in one (2.4%) patient. However, these complications were not life-threatening events. There were no differences in the adverse events following HAIC-FOLFOX-B and HAIC-FOLFIRI-B, except for anorexia and artery occlusion, which were more common among patients who received HAIC-FOLFIRI-B.

## Additional therapy

The treatment course was repeated until tumor progression. In patients assessed for adverse effects or disease progression, additional therapies, such as HAIC with crossover to irinotecan-containing or oxaliplatin-containing chemotherapy or systemic chemotherapy, such as S-1 or immunotherapy, were performed. In this study, additional therapy was administered to 23 patients during the follow-up period. Eleven patients (26.2%) underwent HAIC-FOLFIRI-B, and seven (16.7%) underwent HAIC-FOLFOX-B.

## Discussion

The results of the present study indicated that HAIC-FOLFOX-B and HAIC-FOLFIRI-B were effective in patients who received previous systemic chemotherapy and had been refractory to it. These treatments had a high disease control rate and consequent survival benefits in patients who had undergone previous chemotherapy.

Nowadays, managing colorectal liver metastasis involves the multidisciplinary approach including surgery [25]. Our results are more favorable than those of previous reports. In this study, the median OS period in patients who underwent

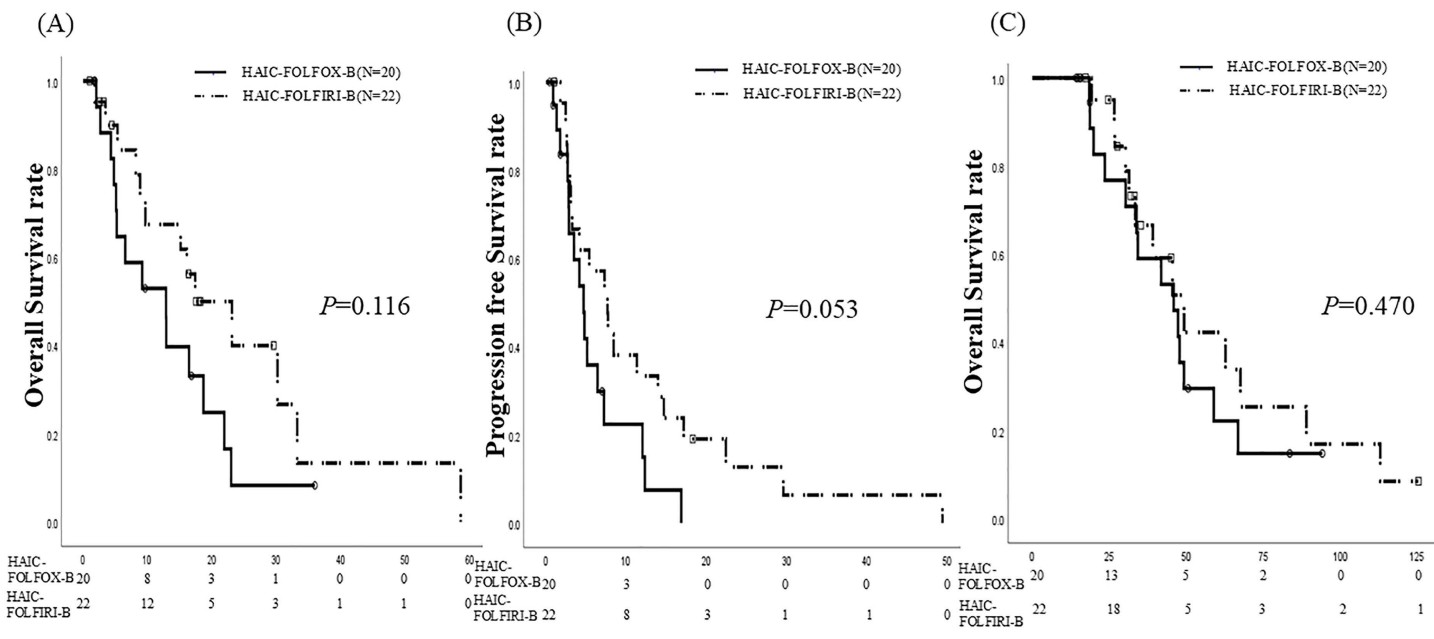

**Fig 2. Analysis of data of 42 patients who underwent HAIC after both systemic oxaliplatin-based and irinotecan-based chemotherapies.** Twenty patients underwent HAIC-FOLFOX-B, and twenty-two underwent HAIC-FOLFIRI-B. **(A)** The OS rates since HAIC-FOLFOX-B initiation were 52.9% at 1 year and 8.3% at 2 years. The median OS period since treatment initiation was 12.9 months (95% CI, 4.892–20.908). The OS rates after HAIC-FOLFIRI-B treatment initiation were 67.5% and 40% at 1 and 2 years, respectively. The median OS periods after treatment initiation was 17.4 months (95% CI, 7.351–27.515). **(B)** The PFS rates since HAIC-FOLFOX-B initiation was 35.4% at 6 months and 22.1% at 1 year. The median PFS period after treatment initiation was 4.7 months (95% CI, 2.957–6.443). The PFS rates after HAIC-FOLFIRI-B initiation were 57.1% and 33.3% at 6 months and 1 year, respectively. The median PFS rate after treatment initiation was 7.7 months (95% CI, 4.178–11.156). There was no significant difference in OS and PFS after treatment initiation between the treatment groups (*P*=0.116, 0.053). **(C)** Among patients who underwent HAIC-FOLFOX-B, the OS rates after disease onset were 59.0% at 3 years and 22.1% at 5 years. The median OS period after disease onset was 45.9 months (95% CI, 28.43–63.37). Among the patients who underwent HAIC-FOLFIRI-B, the OS rates after disease onset were 66.5% and 42.2% at 3 years and 5 years, respectively. The median OS since disease onset was 49.3 months (95% CI, 33.55–65.05). There was no significant difference in OS after disease onset between the treatment groups (*P*=0.470). CI, confidence interval; HAIC-FOLFOX-B, hepatic artery infusion using 5-fluorouracil plus oxaliplatin combination therapy and systemic chemotherapy using leucovorin and bevacizumab; HAIC-FOLFIRI-B, hepatic artery infusion using 5-fluorouracil plus irinotecan combination therapy and systemic chemotherapy using leucovorin and bevacizumab; OS, overall survival; PFS, progression-free survival.

HAIC after both systemic oxaliplatin-based and irinotecan-based chemotherapies (HAIC-FOLFOX-B: median 12.9 months; HAIC-FOLFIRI-B: median, 17.4 months) was longer than that in those who underwent other third-line chemotherapies (regorafenib: median, 6.4 months; and TAS-102: median, 9.0 months) [11,12]. When patients were refractory to systemic standard chemotherapy, HAIC-FOLFOX-B and HAIC-FOLFIRI-B were effective, as shown by the corresponding disease control rates of 70.0% and 86.3%, respectively (Table 2). In addition, HAIC has low toxicity in the whole body. The adverse effects in this study were less frequent than those reported in previous studies (Table 3). Particularly, patients who underwent previous systemic chemotherapy several times had frailty. HAIC is better tolerated than is systemic chemotherapy by patients.

In patients who underwent several systemic chemotherapies assessing adverse effects was important. Although there were almost no differences in the adverse events following HAIC-FOLFOX-B and HAIC-FOLFIRI-B in this study, the patients who underwent HAIC-FOLFIRI-B experienced anorexia and artery occlusion more frequently than did the patients receiving HAIC-FOLFOX-B. The patients underwent systemic FOLFIRI occurred anorexia more frequently than the patients administered systemic FOLFOX [26]. Hepatic arterial occlusion has been reported to be the most frequent cause

**Table 3. Adverse effects of HAIC-FOLFOX-B or HAIC-FOLFIRI-B.**

| Adverse effect | All (N = 42) | | HAIC-FOLFOX-B (n = 20) | | HAIC-FOLFIRI-B (n = 22) | | P-value |
|---|---|---|---|---|---|---|---|
| | Any grade | Grade 3/4 | Any grade | Grade 3/4 | Any grade | Grade 3/4 | |
| Fever | 6 (14.3) | - | 4 (20) | - | 2 (9.1) | - | 0.313 |
| Anemia | 9(21.4) | 7 (16.7) | 4 (20) | 4 (20) | 5 (22.7) | 3 (13.6) | 0.830 |
| AST elevation | 5 (11.9) | - | 2 (10) | - | 3 (13.6) | - | 0.716 |
| General fatigue | 6 (14.3) | - | 2 (10) | - | 4 (18.2) | - | 0.449 |
| Diarrhea | 4 (9.5) | - | 2 (10) | - | 2 (9.1) | - | 0.920 |
| Anorexia | 4 (9.5) | - | - | - | 4 (18.2) | - | 0.045 |
| Infection | 1 (2.4) | 1 (2.4) | - | - | 1 (4.5) | 1 (4.5) | 0.335 |
| Jaundice | 1 (2.4) | - | - | - | 1 (4.5) | - | 0.335 |
| Peripheral neuropathy | 1 (2.4) | - | - | - | 1 (4.5) | - | 0.335 |
| Allergy | 2 (4.8) | 2 (4.8) | 2 (10) | 2 (10) | - | - | 0.129 |
| Artery occlusion | 3 (7.1) | - | 0 (0) | - | 3 (13.6) | - | 0.087 |

Data are expressed as n (%). AST, aspartate aminotransferase; HAIC-FOLFOX-B, hepatic artery infusion using 5-fluorouracil plus oxaliplatin combination therapy and systemic chemotherapy using leucovorin and bevacizumab; HAIC-FOLFIRI-B, hepatic artery infusion using 5-fluorouracil plus irinotecan combination therapy and systemic chemotherapy with leucovorin and bevacizumab.

of interruption of HAIC. [27] Hamada et al. reported that almost 30% of patients who underwent HAIC experienced artery occlusion. [27] Our study showed that 7.1% of the patients underwent HAIC had artery occlusion. Altough this result was favorable, irinotecan might be likely to cause inflammation in the artery. The patients in this study experienced more anemia grade 3/4 than did those in other studies. However many patients underwent several systemic chemotherapies before our treatment in this study. The bone marrow function of such patients might be possibly weakened.

Notably, HAIC, using the same drugs that became ineffective through systemic administration, was effective in this study. Some reports have shown that HAIC using the same drugs was effective in patients with systemic chemotherapy-resistant CRC [28,29] . We reported a similar phenomenon in intrahepatic cholangiocarcinoma [23]. Although the mechanism of this phenomenon is unclear, we present an explanation for the effectiveness of high concentrations of anticancer agents in the liver against resistant tumor cells. Additionally, small amounts of the drug might enter non-tumor tissues throughout the body and reduce drug distribution in organs other than the liver [30]. Although these treatments are suitable only for managing patients with liver metastases, they have high disease control rates and consequent survival benefits.

Various previous studies have evaluated the outcomes of HAIC plus systemic chemotherapy. The data of previous clinical studies are listed in Table 4 [31–41]. Although 42 patients (79.2%) in the present study underwent both systemic oxaliplatin-based and irinotecan-based chemotherapies and were refractory to them, our results are similar to those of other studies. Although it is known that oxaliplatin, irinotecan, and 5-fluorouracil are key drugs for advanced CRC, which drug should be administered through the hepatic artery has not yet been elucidated. We suggest that the administration of high concentrations of anticancer agents is important regarding these treatments. Therefore, bevacizumab and leucovorin were injected intravenously. Although these drugs for intraarterial administration were assumed high concentrations of medication for tumor, there were no difference in effectiveness compared with intravenous administration. HAIC using 5-fluorouracil is more effective in treating liver metastases than is systemic chemotherapy using 5-fluorouracil according to the results of other clinical trials (Table 4). Although HAIC is not recommended for treating patients with advanced CRC, many reports have shown that a combination of HAIC and systemic chemotherapy results in superior tumor control rates compared with systemic chemotherapy alone (Table 4). Although there has been no randomized evidence comparing

**Table 4. Past studies on hepatic artery infusion chemotherapy with systemic therapy for unresectable colorectal cancer liver metastases.**

| Regimen | | Author | Response rate (%) | Median OS since treatment (months) |
|---|---|---|---|---|
| HAIC | Systemic chemotherapy | | | |
| Pirarubicin | 5FU and LV | Fillik et al. [31] | 34.4 | 20 |
| Cisplatin | 5FU | Macini et al. [32] | 52 | 18 |
| Oxaliplatin | 5FU and LV | Ducreux et al. [33] | 64 | 27 |
| 5FU and LV | 5FU and LV | Fiorentini et al. [34] | 47.5 | 20 |
| 5FU and LV | UFT and LV | Tsutsumi et al. [35] | 87.5 | 22 |
| Irinotecan, 5FU, and LV | UFT, and LV | Idelevich et al. [36] | 65 | 36 |
| 5FU | Irinotecan and oxaliplatin | Kemeny et al. [37] | 100 (first line) 85 (prior chemotherapy) | 51 (first line) 35 (prior chemotherapy) |
| 5FU | Irinotecan/oxaliplatin, 5FU, and Bev | D`Angelica et al. [38] | 76 | 38 |
| Irinotecan, oxaliplatin, and 5FU | Cetuximab | Lévi et al. [39] | 40.6 | 25.5 |
| Oxaliplatin | 5FU and LV, or 5FU and Bev, or 5FU and anti EGFR therapy | Lim et al. [40] | 21.3 | 13.5 (first and second line) 8.3 (third and fourth line) |
| 5FU | Irinotecan/oxaliplatin and Bev or 5FU/ LV/irinotecan and Bev | Pak et al. [41] | 73 | 38 |
| 5FU, irinotecan/ oxaliplatin | LV and Bev | Ishii et al. (current study) | 25/4.5 (third or more) | 12.9/17.4 (third or more) |

Bev, bevacizumab; EGFR, epidermal growth factor receptor; HAIC, hepatic artery infusion; LV, leucovorin; OS, overall survival; 5FU, 5-fluorouracil.

HAIC plus systemic chemotherapy with systemic chemotherapy alone in patients with unresectable liver metastasis, a retrospective analysis suggested that adding HAIC to systemic chemotherapy was associated with an improved OS [42]. If extrahepatic metastasis is not advanced, this treatment may be useful as a therapeutic option for use in patients with unresectable CRC liver metastases. According to this study, we found that HAIC-FOLFOX-B was suitable for use in the patients aiming for conversion surgery due to higher response rates. However, there was no difference in the survival between the HAIC-FOLFOX-B and HAIC-FOLFIRI-B groups.

This study has some limitations. First, this was a retrospective analysis performed at a single center only, thus limiting the validity of the results. Second, HAIC requires the expertise of a radiologist; however, not all radiologists can provide the same level of treatment. Therefore, multicenter, prospective, randomized-controlled studies are required to investigate the effectiveness of HAIC-FOLFOX-B and HAIC-FOLFIRI-B. The study design should include equal numbers of systemic chemotherapy and HAIC-FOLFOX-B and HAIC-FOLFIRI-B in patients with unresectable CRC liver metastases after standard chemotherapy.

## Conclusions

HAIC-FOLFOX-B and HAIC-FOLFIRI-B might improve survival in patients who underwent both systemic oxaliplatin-based and irinotecan-based chemotherapies and were refractory to them. If extrahepatic metastasis is not advanced, these treatments may be useful as a therapeutic option for managing patients with unresectable CRC liver metastases after receiving standard chemotherapy.

## Supporting information

**S1 Data. Dataset HAIC CRM1.**
(XLSX)

## Author contributions

**Conceptualization:** Osamu Itano, Satoshi Itano.

**Data curation:** Masatsugu Ishii, Hideki Iwamoto, Yuko Takami, Naomi Okada, Tetsuya Inoue, Satoshi Itano.

**Formal analysis:** Masatsugu Ishii.

**Investigation:** Masatsugu Ishii.

**Methodology:** Masatsugu Ishii, Osamu Itano, Satoshi Itano.

**Project administration:** Satoshi Itano.

**Writing – original draft:** Masatsugu Ishii.

**Writing – review & editing:** Osamu Itano.

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
