## [Decision Letter · Decision Letter 0]

10 Jul 2025

Dear Dr. Itano,

We look forward to receiving your revised manuscript.

Kind regards,

Keun-Yeong Jeong

Academic Editor

PLOS ONE

**Journal Requirements:**

1. When submitting your revision, we need you to address these additional requirements. Please ensure that your manuscript meets PLOS ONE's style requirements, including those for file naming. The PLOS ONE style templates can be found at https://journals.plos.org/plosone/s/file?id=wjVg/PLOSOne_formatting_sample_main_body.pdf and https://journals.plos.org/plosone/s/file?id=ba62/PLOSOne_formatting_sample_title_authors_affiliations.pdf 2. In the online submission form, you indicated that “The datasets used and/or analyzed during the current study are available from the corresponding author on reasonable request”.   All PLOS journals now require all data underlying the findings described in their manuscript to be freely available to other researchers, either a. In a public repository, b. Within the manuscript itself, or c. Uploaded as supplementary information.This policy applies to all data except where public deposition would breach compliance with the protocol approved by your research ethics board. If your data cannot be made publicly available for ethical or legal reasons (e.g., public availability would compromise patient privacy), please explain your reasons on resubmission and your exemption request will be escalated for approval.

Reviewers' comments:

Reviewer's Responses to Questions

**Comments to the Author**

1. Is the manuscript technically sound, and do the data support the conclusions?

Reviewer #1: Partly

Reviewer #2: Yes

2. Has the statistical analysis been performed appropriately and rigorously?

Reviewer #1: Yes

Reviewer #2: Yes

3. Have the authors made all data underlying the findings in their manuscript fully available?

Reviewer #1: Yes

Reviewer #2: No

4. Is the manuscript presented in an intelligible fashion and written in standard English?

Reviewer #1: Yes

Reviewer #2: Yes

**Reviewer #1: ** Main issues

1.Inadequate Patient Allocation Justification No explanation for the non-randomized assignment to HAIC-FOLFOX-B (n=20) vs. HAIC-FOLFIRI-B (n=22). Clarify selection criteria (e.g., physician discretion, tumor characteristics) to address potential selection bias.

2.Contradictory Efficacy Conclusions Abstract claims HAIC-FOLFOX-B is effective (RR 25%), yet results show inferior median OS (12.9 months) vs. HAIC-FOLFIRI-B (17.4 months). Reconcile this discrepancy and clarify clinical significance given non-significant p-values (OS: p=0.116; PFS: p=0.053).

3.Incomplete Safety Reporting Table 3 shows significantly higher anorexia in HAIC-FOLFIRI-B (18.2% vs. 0%, p=0.045) and artery occlusion (13.6% vs. 0%), yet these are not discussed. Expand safety analysis to address toxicity trade-offs between regimens.

4.Unclear Definition of "Refractory" Status Lack of standardized criteria for prior chemotherapy failure (e.g., RECIST progression, cycles completed). Specify how "refractoriness" was determined to ensure patient homogeneity.

**Reviewer #2:**  This manuscript addresses an important clinical question about treatment options for patients with unresectable colorectal liver metastases refractory to standard systemic chemotherapy. The authors present a retrospective cohort study comparing two hepatic arterial infusion (HAIC) regimens combined with systemic bevacizumab. The topic is clinically relevant and the data are potentially useful for clinicians managing a difficult-to-treat patient population.

While the study is interesting, some major and minor issues should be addressed to improve its clarity, rigor, and presentation.

1. The manuscript mentions that all patients had previously failed systemic oxaliplatin-based and irinotecan-based therapies, but Table 1 shows heterogeneity in prior treatments (ranging from 2 to 7 prior lines). Please clarify how “refractory” was defined.

2. The p-values reported for OS and PFS between the two HAIC regimens are >0.05, yet the discussion sometimes implies superiority of one regimen over another.

3. Some toxicities (e.g., 16.7% grade 3–4 anemia, hepatic artery occlusion) are not trivial. Please clarify.

4. Figures are not visible in the PDF.

5. Table 4 is extensive but difficult to follow.

6. The abstract does not specify that the study was retrospective or single-center.

**Do you want your identity to be public for this peer review?** For information about this choice, including consent withdrawal, please see our Privacy Policy

Reviewer #1: No

Reviewer #2: No

---

## [Author Response · Author response to Decision Letter 1]

4 Aug 2025

Comments to the Author

Reviewer #1: Main issues

1.Inadequate Patient Allocation Justification No explanation for the non-randomized assignment to HAIC-FOLFOX-B (n=20) vs. HAIC-FOLFIRI-B (n=22). Clarify selection criteria (e.g., physician discretion, tumor characteristics) to address potential selection bias.

Response:

We thank you for the suggestion. Accordingly, we explained the reason in the revised manuscript.

Page 9 line 129-130

Regarding patient allocation, the physician in charge considered the previously administered regimen and decided the regimen to be selected.

Comments to the Author

2. Contradictory Efficacy Conclusions Abstract claims HAIC-FOLFOX-B is effective (RR 25%), yet results show inferior median OS (12.9 months) vs. HAIC-FOLFIRI-B (17.4 months). Reconcile this discrepancy and clarify clinical significance given non-significant p-values (OS: p=0.116; PFS: p=0.053).

Response:

We appreciate and agree with you on this comment. Abstracts showed HAIC-FOLFOX-B/ FOLFIRI-B regimens might be effective. We modified it as below.

Page 4 lines 47-49

HAIC FOLFOX-B and FOLFIRI-B regimens might be effective therapeutic options in patients with unresectable colorectal liver metastases refractory to standard systemic chemotherapy.

Comments to the Author

3.Incomplete Safety Reporting Table 3 shows significantly higher anorexia in HAIC-FOLFIRI-B (18.2% vs. 0%, p=0.045) and artery occlusion (13.6% vs. 0%), yet these are not discussed. Expand safety analysis to address toxicity trade-offs between regimens.

Response:

We added a text to the Discussion section to expand it.

Pages 22-23 lines 272-285

In patients who underwent several systemic chemotherapies, assessing adverse effects was important. Although there were almost no differences in the adverse events following HAIC-FOLFOX-B and HAIC-FOLFIRI-B in this study, the patients who underwent HAIC-FOLFIRI-B experienced anorexia and artery occlusion more frequently than did the patients receiving HAIC-FOLFOX-B. The patients who underwent systemic FOLFIRI experienced anorexia more frequently than did the patients administered systemic FOLFOX [26]. Hepatic arterial occlusion has been reported to be the most frequent cause of interruption of HAIC [27]. Hamada et al. reported that almost 30% of patients who underwent HAIC experienced artery occlusion. [27] Our study showed that 7.1% of the patients who underwent HAIC had artery occlusion. Although this result was favorable, irinotecan might be likely to cause inflammation in the artery. The patients in this study experienced more anemia grade 3/4 than did those in other studies. However, many patients underwent several systemic chemotherapies before our treatment in this study. The bone marrow function of such patients might be possibly weakened.

Comments to the Author

4.Unclear Definition of "Refractory" Status Lack of standardized criteria for prior chemotherapy failure (e.g., RECIST progression, cycles completed). Specify how "refractoriness" was determined to ensure patient homogeneity.

Response:

Thank you for your suggestions. We added the explanation to the revised manuscript.

Page 8 lines 105-107

All of these patients were refractory to both oxaliplatin-based and irinotecan-based chemotherapies. Refractory patients were defined as those who could not receive either regimen again due to adverse effects or disease progression. 

Reviewer #2: This manuscript addresses an important clinical question about treatment options for patients with unresectable colorectal liver metastases refractory to standard systemic chemotherapy. The authors present a retrospective cohort study comparing two hepatic arterial infusion (HAIC) regimens combined with systemic bevacizumab. The topic is clinically relevant and the data are potentially useful for clinicians managing a difficult-to-treat patient population.

While the study is interesting, some major and minor issues should be addressed to improve its clarity, rigor, and presentation.

Comments to the Author

1. The manuscript mentions that all patients had previously failed systemic oxaliplatin-based and irinotecan-based therapies, but Table 1 shows heterogeneity in prior treatments (ranging from 2 to 7 prior lines). Please clarify how “refractory” was defined.

Response:

We added the explanation to the revised manuscript.

Page 8 lines 105-107

All of these patients were refractory to both oxaliplatin-based and irinotecan-based chemotherapies. Refractory patients were defined as those who could not receive either regimen again due to adverse effects or disease progression.

Comments to the Author

2. The p-values reported for OS and PFS between the two HAIC regimens are >0.05, yet the discussion sometimes implies superiority of one regimen over another.

Response:

Thank you for your comment. We added an explanation to the revised text.

Page 24-25 lines 317-320

According to this study, we found that HAIC-FOLFOX-B was suitable for use in the patients aiming for conversion surgery due to higher response rates. However, there was no difference in the survival between the HAIC-FOLFOX-B and HAIC-FOLFIRI-B groups.

Comments to the Author

3. Some toxicities (e.g., 16.7% grade 3–4 anemia, hepatic artery occlusion) are not trivial. Please clarify.

Response:

We appreciate and agree with you about the comment. We added an explanation to the Discussion section.

Page 22-23 lines 278-285

Hepatic arterial occlusion has been reported to be the most frequent cause of interruption of HAIC. [27] Hamada et al. reported that almost 30% of patients who underwent HAIC experienced artery occlusion. [27] Our study showed that 7.1% of the patients underwent HAIC had artery occlusion. Altough this result was favorable, irinotecan might be likely to cause inflammation in the artery. The patients in this study experienced more anemia grade 3/4 than did those in other studies. However, many patients underwent several systemic chemotherapies before our treatment in this study. The bone marrow function of such patients might be possibly weakened.

Comments to the Author

4. Figures are not visible in the PDF.

Response:

We added the figures again.

Comments to the Author

5. Table 4 is extensive but difficult to follow.

Response:

We omitted some of Table's lines.

Comments to the Author

6. The abstract does not specify that the study was retrospective or single-center.

Response:

We added a sentence to the abstract to specify the study design.

Page 3 lines 31-32

This study was a retrospective analysis conducted at a single center.

---

## [Decision Letter · Decision Letter 1]

22 Oct 2025

Hepatic arterial infusion is effective in patients with unresectable colorectal liver metastases refractory to standard systemic chemotherapy: A retrospective cohort study

PONE-D-25-22185R1

Dear Dr. Itano,

We’re pleased to inform you that your manuscript has been judged scientifically suitable for publication and will be formally accepted for publication once it meets all outstanding technical requirements.

Kind regards,

Keun-Yeong Jeong

Academic Editor

PLOS ONE

Additional Editor Comments (optional):

Reviewers' comments:

Reviewer's Responses to Questions

**Comments to the Author**

Reviewer #2: All comments have been addressed

2. Is the manuscript technically sound, and do the data support the conclusions?

Reviewer #2: Yes

3. Has the statistical analysis been performed appropriately and rigorously?

Reviewer #2: Yes

4. Have the authors made all data underlying the findings in their manuscript fully available?

Reviewer #2: Yes

5. Is the manuscript presented in an intelligible fashion and written in standard English?

Reviewer #2: Yes

Reviewer #2: (No Response)

**Do you want your identity to be public for this peer review?** For information about this choice, including consent withdrawal, please see our Privacy Policy

Reviewer #2: No

---

## [Editor Report · Acceptance letter]

PONE-D-25-22185R1

PLOS ONE

Dear Dr. Itano,

I'm pleased to inform you that your manuscript has been deemed suitable for publication in PLOS ONE. Congratulations! Your manuscript is now being handed over to our production team.

Kind regards,

on behalf of

Dr. Keun-Yeong Jeong

Academic Editor

PLOS ONE